# Studies on Cistanches Herba: A Bibliometric Analysis

**DOI:** 10.3390/plants12051098

**Published:** 2023-03-01

**Authors:** Longjiang Wu, Tian Xiang, Chen Chen, Murtala Bindawa Isah, Xiaoying Zhang

**Affiliations:** 1Chinese-German Joint Laboratory for Natural Product Research, Qinba State Key Laboratory of Biological Resources and Ecological Environment, Shaanxi University of Technology, Hanzhong 723001, China; 2Department of Biochemistry, Faculty of Natural and Applied Sciences, Umaru Musa Yar’adua University Katsina, P.M.B. 2218, Katsina 820102, Nigeria; 3Biomedical Research and Training Centre, Yobe State University, P.M.B. 1144, Damaturu 600213, Nigeria; 4Centre of Molecular and Environmental Biology (CBMA), Department of Biology, University of Minho, Campus de Gualtar, 4710-057 Braga, Portugal; 5Department of Biomedical Sciences, Ontario Veterinary College, University of Guelph, Guelph, ON N1G 2W1, Canada

**Keywords:** *Cistanche*, CiteSpace, edible and medicinal plant, bibliometric analysis, Web of Science

## Abstract

As a famous tonic herb, Cistanches Herba is known for its broad medicinal functions, especially its hormone balancing, anti-aging, anti-dementia, anti-tumor, anti-oxidative, neuroprotective, and hepatoprotective effects. This study aims to provide a comprehensive bibliometric analysis of studies on *Cistanche* and to identify research hotspots and frontier topics on the genus. Based on the metrological analysis software CiteSpace, 443 *Cistanche* related papers were quantitatively reviewed. The results indicate that 330 institutions from 46 countries have publications in this field. China was the leading country in terms of research importance and number of publication (335 articles). In the past decades, studies on *Cistanche* have mainly focused on its rich active substances and pharmacological effects. Although the research trend shows that *Cistanche* has grown from an endangered species to an important industrial plant, its breeding and cultivation continue to be important areas for research. In the future, the application of *Cistanche* species as functional foods may be a new research trend. In addition, active collaborations among researchers, institutions, and countries are expected.

## 1. Introduction

*Cistanche* Hoffmg. et Link (*Cistanche*) is a genus of the family Orobanchaceae, which includes 27 species accepted in The Plant List (http://www.theplantlist.org, accessed on 22 November 2022). *Cistanche* species are obligate parasitic plants growing on the roots of sand-fixing plants and are mainly distributed in arid and semiarid regions of Asia, the Iberian Peninsula in Europe, and North Africa [1,2]. The succulent stem is the main medicinal part of *Cistanche* Herba plants, which is usually derived from *Cistanche deserticola* Y.C.Ma, *Cistanche tubulosa* (Schrenk) Wight ex Hook.f., *Cistanche salsa* (C.A.Mey.) Beck and *Cistanche sinensis* Beck [3]. *C. deserticola* and *C. tubulosa* are the most commonly used and have been documented in Chinese Pharmacopoeia [4]. *Cistanche deserticola*, *C. tubulosa* and *C. salsa* are included in Japanese Pharmacopoeia as the crude drug “Nikujuyou” [5]. They are often used to treat kidney deficiencies as well as blood and kidney disorders [6].

Already, more than 200 compounds have been identified from *Cistanche* Herb plants, with the major components including phenylethanoid glycosides (PhGs), oligosaccharides, polysaccharides, essential oils, cistanosides, iridoids, lignans, and alditols [7,8]. Five PhGs, namely echinacoside, cistanoside A, acteoside, isoacteoside, and 2′-acetylacteoside, were separated and purified using high-speed counter-current chromatography (HSCCC) [9]. Then, structures of all five PhGs were characterized with liquid chromatography electrospray ionization mass spectrometry (LC-ESI-MSn) and nuclear magnetic resonance spectroscopy (NMR). In addition, the combination of HSCCC with the high-performance liquid chromatographic method with evaporative light-scattering detection (HPLC-ELSD), high-resolution mass spectrometry (HR-MS), and NMR demonstrated efficiency in the separation and characterization of the chemical constituents in *C. deserticola* [10]. Recent studies demonstrated that the active ingredients of *C. tubulosa* can be better purified using novel adsorption materials (molecularly imprinted polymers, mesoporous carbon, etc.) [11,12]. Presently, scholars are dedicated to the study of the pharmacologically active ingredients and related mechanisms of action of *Cistanche* through advanced analytical identification techniques, such as ultra-performance liquid chromatography coupled with quadrupole time-of-flight tandem mass spectrometry (UPLC-Q-TOF-MS). Cui et al. developed an UPLC-Q-TOF-MS technique to characterize the metabolites of acteoside produced by human or rat intestinal bacteria or rat intestinal enzymes, and analyzed the similarities and differences of metabolic pathways and processes [13,14]. Li et al. screened and characterized the metabolites of *C. tubulosa* extract using UPLC-Q-TOF-MS in healthy rats and depressed chronic obstructive pulmonary diseased rats [15]. Their research showed that the metabolic capacity of the normal rat gut microbiota to produce secondary glycosides and glycans was significantly stronger than the gut microbiota of the depressed rats, which formed the basis for better knowledge on the metabolic process and therapeutic mechanism of the antidepressant effect of *C. tubulosa* extract. These high-efficiency and -accuracy isolation and identification techniques will not only facilitate the clinical study of the active ingredients, but can also provide a strong guarantee for the quality control of *Cistanche* products.

Modern pharmacological studies have demonstrated a wide range of pharmacological effects of *Cistanche* Herba plants, such as immunomodulatory [16], hormonal balancing [17], anti-inflammatory [18], hepatoprotective [19], anti-neurodegenerative disease [20,21], and anti-osteoporotic [22,23]. The main pharmacological activities of Cistanches Herba are summarized in Table 1. PhGs are a well-known compound group in *Cistanche* [24]. Among them, the biological activities of acteoside, echinacoside, and cistanoside have been extensively studied. Acteoside, also known as verbascoside, has demonstrated neuroprotection and memory improvement activities [25,26] by reducing the apoptosis of neurons and loss of memory associated with the combined induction of D-galactose and AlCl_3_ in a mouse model of aging. Echinacoside could ameliorate the cognitive dysfunction developed in a Alzheimer’s disease-like rat model [20], and could be a druggable compound in improving sperm quality, reducing testicular toxicity in a bisphenol A–induced reproductive damage model in rats via up-regulation of steroidogenesis enzymes [17,27] based on a systemic effect of blocking androgen receptor activity in the hypothalamus [28]. One study suggested cistanoside could ameliorate hypoxia-induced male reproductive damage via suppression of oxidative stress [29]. A recent animal study suggested that the PhG-enriched extract from *C. deserticola* was effective in protecting myocardial cells and reducing cell apoptosis and injury by reducing the expression of the endoplasmic reticulum stress (ERS)-associated apoptotic factors [30]. *Cistanche deserticola* PhGs exhibited good hepatoprotective effects by stimulating hepatocyte proliferation and regeneration and decreasing the process of inflammation, oxidative stress, and apoptosis [31]. *Cistanche tubulosa* PhGs may be potential candidates as therapeutic agents for hepatocellular carcinoma (HCC) [32,33], inhibiting the growth of H22 cells and causing apoptosis through signaling pathways in vitro and in vivo. In recent years, the pharmacological activities of *C. deserticola* polysaccharides (CDP) have attracted increasing attention. CDP is capable of promoting melanin synthesis and preventing oxidative stress damage to melanocytes [34] and is likely a new source of herbal medicine in treating depigmentation diseases. CDP can also be used as a safe and effective immunization adjuvant for the induction of humoral and cellular immunity. Water-extractable CDP could modulate immune responses both in vitro and in vivo [35]. Meanwhile, CDP exhibited a good protective effect on ovariectomized (OVX)-induced osteoporosis by repressing osteoclast activity and function [36]. Another study demonstrated that CDP could potentially promote the prevention of osteoporosis in senescence accelerated mouse prone 6 (SAMP6) mice via activation of the Wnt/β-catenin signaling pathway [37]. Its extensive pharmacological activity is the reason why CDP has been increasingly studied and reported, and its potential application in the future is expected to be expanded, especially in clinical aspects. In addition, the neuroprotective effects of *Cistanche* are an ongoing subject of significant research, with an increasing number of studies revealing its great potential in the treatment of neurological dysfunction, including Alzheimer’s and Parkinson’s diseases. In 2016, Gu et al. conducted a comprehensive review of the neuropharmacological effects and the mechanism of neuroprotective activity of Cistanches Herba extracts [38], which provided an updated view on further studies in this area. The great neuroprotective effects supported the clinical importance of *Cistanche* and provided a potential basis for new drugs to treat neurological disorders. However, the development and discovery of a new *Cistanche-*based drug requires more detailed studies of the pharmacological mechanisms, pharmacokinetics, and clinical applications of its products, particularly at the molecular level. Further studies are also required to improve bioavailability and to clarify the mechanism of action of absorption in the human body.

**Table 1 plants-12-01098-t001:** Pharmacological effect and mechanism of action of Cistanches Herba.

Pharmacological Effect	Compounds	Mechanism of Action	References
Neuroprotection effect	Echinacoside, acteoside	Prohibiting the apoptosis of neurons	[39,40,41]
Bone metabolism regulation effect	Echinacoside, acteoside, *C. deserticola* polysaccharides (CDP), cistanoside A	Suppressing NF-κB and c-Fos pathways and downregulating NF-κB ligand receptor activator (RANKL)	[42,43,44]
Hepatoprotection activity	Phenylethanoid glycosides (PhGs), CDP	Reducing levels of triglycerides, malondialdehyde (MDA), and low-density lipoproteins	[45,46]
Anti-inflammation and immunoregulation activity	PhGs, CDP, oligosaccharides, cistanoside K, and tubuloside B	Promoting proliferation of T cells, reducing inflammatory hyperplastic polyps, helicobacter infection, and nitric oxide production	[18,35,47]
Antiaging effect	Echinacoside, oligosaccharides, CDP	Suppressing reactive oxygen species production, regulating Nrf2/heme oxygenase-1-dependent antioxidative pathway	[34,48,49]
Antifatigue effect	PhGs, CDP, acteoside	Delaying lactic acid accumulation, suppressing 5-hydroxytryptamine synthesis increase	[50,51,52]
Reproductive regulation	Echinacoside, PhGs	Decreasing MDA content, inhibiting the transformation of androgen receptor	[28,53]
Anticancer effect	Echinacoside, PhGs, CDP	Reducing H22 tumor-bearing liver injury and inhibiting the growth of K562 leukemia cells	[54,55]

Historically, *Cistanche* Herba has been a valuable medicinal plant, available only through wild resources. *Cistanche* can only be reproduced by seeds, and as obligate parasites, the root exudates of their hosts play a dominant role in seed germination and haustorium production [24]. A study showed that *C. deserticola* mature seeds contained non-differentiated embryos that were physiologically dormant and only responded to germination stimulants if physiological dormancy was broken [56]; however, the mechanism underlying the breaking of dormancy and germination is yet to be understood. This signifies the need for more research into this phenomenon. Compared with *Haloxylon ammodendron*, *Atriplex canescens* is the more preferable host for the industrial production of *C. deserticola* [57]. A major contribution to this is the high biomass and wide ecological adaptation of *A. canescens*; *C. deserticola,* which is parasitic on *A. canescens,* produces higher concentration of active ingredients than when parasitized on *H. ammodendron*. In recent years, benefiting from the large-scale breeding and cultivation, the price of *Cistanche* herb has dropped significantly, though it has retained a semi-wild quality [58,59], which has laid the foundation for its popularization and application, especially for functional food application. Researchers have conducted multiple attempts to study *Cistanche* as a functional food. The mixture of *Cistanche* PhGs and gardenia yellow pigment showed good anti-hypoxia and anti-fatigue activity [60], which may serve as a strategic and functional product for the improvement of low-oxygen exercise fatigue. The functional and physiological properties of *C. deserticola* can be used in the development of composite rice products. A good-tasting composite rice product was produced by utilizing ground rice, potato flakes, and *C. deserticola* [61]. The stem of *C. deserticola* had promising anti-xanthine oxidase activity [62], as a functional food with strong anti-gout activity. The acteoside metabolites in *Cistanche* tea were reported as a good active substance for liver protection [63], thus *Cistanche* tea is available for development as a functional beverage against liver injury. *Cistanche* products (tea, wine, etc.) are increasingly being registered on the market. *Cistanche* has grown from an endangered species to a major brand of Chinese herbal medicine, with over 20 billion Yuan (RMB) of related industrial output [24]. *Cistanche deserticola* has been recognized as a “medicine food homology” (MFH) in China since 2018 [64]. With the inclusion of *Cistanche* in the list of edible herbal medicines, *Cistanche* and its related products will have a greater opportunity for development as a recommended major health product.

This study aims to employ CiteSpace to analyze the status and to explore the hot spots and trends of *Cistanche* study, as a timely update of our previous review [6], after the rapid development of this field in the past decade. CiteSpace is a Java-based scientometric software for the analysis and visualization of scientific literature and data [65].

## 2. Results

### 2.1. Analysis of Important Literature

The collection of all previous publications is the foundation of the frontier research in a subject. There were 799 co-citations among 443 publications, resulting in 3037 links (Figure 1). It should be noted that the citation frequency in this article was limited to the mutual citation among these 443 articles so that the specific citation frequency would be different from the statistics in WOS. The top ten co-cited publications were mainly published after 2010, while the works with the highest centrality were published between 2001 and 2010, with the article of Xuefei Tian et al. published in the *Journal of Ethnopharmacology* in 2005 ranked first. The most cited article was published by Zhiming Li in 2016 in the *Journal of Chromatography A*, entitled “CH (Rou Cong-Rong): One of the Best Pharmaceutical Gifts of Traditional Chinese Medicine”.

The analysis of reviews is helpful for understanding the main discoveries, emerging topics, and views of experts on a subject over a specific period of time. The major findings and the gaps identified in 10 highly cited reviews on *Cistanche* study are summarized in Table 2. These reviews provide a strong conceptual appeal and lay the foundation for future studies on the phytochemistry and quality control of *Cistanche*.

**Table 2 plants-12-01098-t002:** Summary of major review articles published on *Cistanche* (1980–2022).

Topic Reviewed	Major Findings	Gaps Identified	Reference
Chemical constituents and analytical methods	PhGs are the main active components, and HPLC was the most widely used method	Comparative analysis of multi-component fingerprints was lacking	[7]
Botany, traditional uses, phytochemistry, and pharmacology	The traditional use has been confirmed by modern pharmacological research	Studies on monomer components were lacking; quality control methods were not unified	[6]
Biological characteristics, chemical constituents, and pharmacological activities	Cistanches Herba has great potential as a drug candidate for the treatment of a variety of diseases	Lack of study on the monomer components and clinical practice	[66]
Neuropharmacological and neuroprotective mechanisms	Most of the neuropharmacological effects are closely related to antioxidant activity	The mechanisms of human absorption and bioavailability are unresolved	[38]
Taxonomy, distribution, biological functions, and molecular mechanisms	Various Cistanches Herba products and their derivatives are widely used	Compounds with pharmacological activity lack more in-depth studies	[67]
Anti-aging effects or anti-aging-related effects	There are significant therapeutic and economic advantages to the development of new drugs	The exact compounds responsible for the observed pharmacological effects remain unclear	[68]
Distribution, preparation processes, pharmacokinetics, and therapeutic uses of echinacoside	Echinacoside shows a high degree of positive activity in neurological diseases	Clinical trials on the safety and the druggability of echinacoside are lacking	[69]
Phytochemistry, pharmacology, concoction, toxicity, and safety	Most drugs focus only on phenotypic analysis, hindering the development of new drugs	Research in molecular biology, bioinformatics and chemical biotechnology remains inadequate	[8]
Distribution and cultivation, phytochemistry, pharmacology, metabolism, and product development	Cistanche has grown to become a big brand industry from an endangered species	Some obvious bottlenecks in parasitic mechanisms, production development, and environmental balance	[24]
Neuroprotective effects of echinacoside	Echinacoside can be used as an effective and safe substance in the treatment of neurodegenerative diseases	Clinical evidence is still lacking	[70]

### 2.2. Annual Number of Publications

From 1980–2022, 443 *Cistanche*-themed publications were indexed on WOSCC, including 417 articles (94.13%), 21 reviews (4.74%), and 5 book chapters (1.13%). According to the WOS citation report, the cumulative number of citations to these publications was 7958, at an average of 17.96 per paper. Annual publication statistics showed that the number of publications increased in three stages (Figure 2). After a slow rise between 1980 and 2003 (0 to 5 publications per year), the yearly number of published papers increased significantly in the following decade, to reach 19 papers in 2012. Since then, the number of publications has increased steadily and is expected to exceed 50 in 2022. Accordingly, the number of *Cistanche*-related publications is at an all-time high, with a clear upwards trajectory.

### 2.3. Main Research Forces

#### 2.3.1. Major Countries

From 1980 to 2022, a total of 46 countries/regions published *Cistanche*-themed papers. In general, there was a large gap in the research on *Cistanche* across countries (Figure 3A). European and Asian countries/regions predominate in *Cistanche* study. China (including Taiwan) was the leading country in terms of publications, with 301, far above Japan (33) and the United States (15) (Figure 3B). Countries with more publications have established cooperative relations and formed a larger network. It is noteworthy that only a few countries have published articles in recent years, but the number of total articles has been increasing. Some countries produced publications on *Cistanche* only in the early years.

#### 2.3.2. Main Institutions

A key aspect of integration into the global research community occurs through international scientific collaborations [71]. A total of 330 institutions were involved in *Cistanche* study, with 443 links established (Figure 4). As a major contributor, China contained eight out of the top ten institutions exploring this field. Peking University, Chinese Academy of Sciences, Beijing University of Chinese Medicine, and Chinese Academy of Medical Sciences made the largest contributions. Their studies covered cultivation and breeding, medicinal components, pharmacological activities, and quality monitoring of *Cistanche* [57,72,73,74]. Kinki University and Kyoto Pharmaceutical University were the leading institutions in Japan. Most of their papers were published after 2006 and focused extensively on studies of the hepatoprotective and vasodilatory effects of *C. tubulosa* [75,76]. Researchers from the United States and Poland mainly focused on the phylogeny and taxonomy of *Cistanche* [77,78,79]. China accounts for nine of the top ten funding institutions, of which 153 projects were funded by the National Natural Science Foundation of China (NSFC), accounting for 34.76% of the total (Table 3). More than half of all funding institutions were from China, with a wide range of funding categories, covering basic research, quality control, and other areas.

#### 2.3.3. Main Authors

Prolific authors usually have stable collaborations among one another (Figure 5). A high intermediary centrality (>0.1) of the nodes representing the top three authors demonstrates that these authors are more influential and collaborate with others more frequently. Noteworthy was the fact that 21 researchers of the top 25 authors were from China and the other 4 were from Japan. Pengfei Tu from Peking university published the most papers, with 50, accounting for 11.27% of the total, far higher than Yuxin Wang from China Pharmaceutical University, who ranked second (34 publications).

### 2.4. Source Analysis of Journals

As depicted in Figure 6, *Cistanche-*related publications were widely disseminated in different journals. The top-ranked journal by citation counts was *Journal of Ethnopharmacology,* with 199 citations, followed by *Journal of Chromatography A* (151 citations), *Planta Medica* (140 citations), *Biological & Pharmaceutical Bulletin* (128 citations), and *Chemical & Pharmaceutical Bulletin* (118 citations). *Phytochemistry*, *Food Chemistry*, *Molecules*, *Biomedicine & Pharmacotherapy*, and *Journal of Functional Foods* produced a high citation burst, indicating their critical role in this field.

### 2.5. Hotspot Analysis

#### 2.5.1. Outbreak Word Analysis

“Burst words” are keywords appearing frequently and reflect the evolution of the hot topics and relevant disciplines during a period of time. Figure 7 shows the top 25 burst keywords in this field from 1980–2022. “Extract” (2016–2020) was the strongest citation burst keyword, followed by “mice” (2016–2018), and “identification” (2017–2019). Beyond the keywords noted above, bioactive constituent (2006–2014), Chinese natural medicine (2006–2014), gymnadenia conopsea (2006–2008), neuronal cell (2016–2018), pharmacokinetics (2016–2017), metabolite (2018–2019), mechanism (2018–2020), and expression (2020–2022) all have high intensity, which suggest that they were among of the main topics discussed at different stages of *Cistanche* study.

#### 2.5.2. Keyword Analysis

The high-frequency keywords reflected the hot issues, while high-centricity keywords presented the important and influential topics of *Cistanche*-related study. The keyword co-occurrence and clustering are presented in Figure 8 and Table 4. The visual network diagram showed 562 keywords in 443 articles, of which 3 keywords appear 70 times or more and 11 keywords appear 20 times or more. Those keywords with the biggest nodes, density, and frequency were *C. deserticola*, phenylethanoid glycoside, *C. tubulosa*, constituent, and echinacoside (139, 111, 71, 37, and 34 occurrences, respectively) (Figure 8). In addition, concerning other keywords, some are related to *Cistanche* study on diseases, disease conditions and treatments like apoptosis, in vitro, Parkinson’s disease, Alzheimer’s disease, oxidative stress, antioxidant activity, and inflammation. Several descriptive terms, for instance, model, pathway, expression, accumulation, stress, growth, evolution, diversity, quality, and assay, correspond to the attention of scholars on different aspects of *Cistanche* study.

The mean silhouette was used to evaluate the clusters. Mean silhouette > 0.50 means the members of the cluster have some similarity and homogeneity; if it is higher than 0.70, the cluster is highly efficient and convincing. As shown in Table 4, the CiteSpace-based co-occurring keywords have been categorized into eight main clusters; the silhouette values of all cluster algorithms exceeded 0.70, indicating that the results were reliable and convincing: #0 “Parkinsons disease”, #1 “Cistanoside a”, #2 “Antioxidant activity”, #3 “Phylogeny”, and #4 Phenylalanine ammonia lyase (PAL) represent the focus themes of the *Cistanche* study. Keyword analysis demonstrated that studies on *Cistanche* have mainly focused on their abundant active substances and pharmacological effects, especially for *C. deserticola* and *C. tubulosa* (Figure 8, Table 4).

## 3. Discussion

In the last decade, studies on *Cistanche* have increased rapidly, as evident in the upward trend in the number of annual publications and frequency of citations (Figure 2). A systematic and comprehensive analysis of *Cistanche*-related publications is essential. Bibliometric analysis overcomes the subjective element in traditional reviews and allows quantitative mining of the knowledge structure, research hotspots, and new discoveries in certain scientific fields [80]. Our previous article [6], as a first attempt, comprehensively discussed and summarized the botany, traditional uses, phytochemistry, and pharmacology of *Cistanche*. To provide a more comprehensive and objective guide for further studies on *Cistanche*, this work provides a bibiometric perspective with a quantitative and visualization analysis of the studies on *Cistanche*. On this basis, we present the status of *Cistanche* study and reveal the research hotspots and future trends.

Analysis of the metrological results demonstrated that the institutions and authors from East Asian and European countries have higher visibility and greater contributions in *Cistanche* study (Figure 3, Figure 4 and Figure 5). Chinese academic institutions stood out with dozens of institutes involved in the study of *Cistanche*. In terms of academic exchange and collaboration, China often cooperated with Eastern Asian countries, while collaboration with other countries was infrequent (Figure 3A). Researchers from the United States had less international collaboration, except limited activities with Spain, Austria, Nigeria, and Switzerland. Collaborations within European countries were also weak. In addition, adequate financial support is an important prerequisite for conducting *Cistanche*-related study; in this regard, China and Japan have given a higher priority (Table 3). Collaborations among countries and institutions in *Cistanche* study need to be strengthened further, which will help to provide a panoramic understanding and to promote an in-depth understanding of this medicinally important plant.

With extended efforts, comprehensive and in-depth study on *Cistanche* has been obtained, from basic taxonomic research to germplasm resource exploitation, and from isolation of bioactive components to elucidation of their pharmacological activities and their molecular mechanisms. Keywords can represent topics of interest in a given field. Outburst words revealed changes in the focus of the study over the years (Figure 7). Keywords co-occurring and clustering analysis indicated the current research trends include the abundant active substances and their pharmacological effects (Figure 8, Table 4). *Cistanche deserticola* and *C. tubulosa* have been highlighted in *Cistanche* study. Apoptosis, in vitro, Parkinson’s and Alzheimer’s diseases, oxidative stress, antioxidant activity, and inflammation were the topics being intensively investigated and reported.

Earlier researchers relied on physical qualities in differentiating the species, which was difficult and often inaccurate [81]. Molecular phylogenetic analysis provides a good way to distinguish the intrinsic differences between species. The phylogenetic relationships of *C. deserticola*, *C. salsa,* and *C. tubulosa* were revealed by the work of Tu et al. using DNA sequencing [82]. The NJ tree constructed from the sequence data showed *C. deserticola* was closely related to *C. salsa*, and *C. tubulosa* was one of the outgroups. A study on molecular phylogenetic analysis with intranuclear transcribed spacer (ITS) sequences including all members of the Orobanche family was performed by Schneeweiss et al. [77]. This was the first time that *Cistanche* was identified as a major lineage in the Orobanchaceae family by molecular phylogenetic analysis. But the relationships of the *Cistanche* genus and the clade, including the rest of the genus has not been identified. A recent study inferred phylogenetic relationships within *Cistanche* using a molecular phylogenetic approach [83], taking taxonomically and geographically extensive sampling, using maximum parsimony, maximum likelihood, and Bayesian inference combined matrix analysis. *Cistanche* was classified into four well-supported and geographically distinct clades. However, some species still need to be included in molecular phylogenetic studies. Based on the molecular and morphological data, the species diversity of *Cistanche* needs further study.

The historical progress on the isolation and purification of high purity bioactive components from *Cistanche* suggests that researchers have continually adapted new methods for the study of *Cistanche*, yielding better results. This is critical for the accurate investigation of the pharmacological activities of *Cistanche*. At the early stage, HSCCC was successfully used as a major separation method to isolate numerous active components of PhGs from *Cistanche* [9,10,84,85]. The UHPLC/ESI-QTOF-MS/MS method established by Han et al. successfully identified or preliminarily identified a total of 13 PhGs in the *C. deserticola* crude extract [86], which provided an important basis for the rapid prediction of chemical composition and quality of the plant. A systematic phytochemical study was carried out to determine the chemical constituents of cultivated *Cistanche* in the Tarim Desert [87]. The compounds were separated by silica gel, Sephadex LH-20, MCI gel, ODS column chromatography, and semi-preparative HPLC. After the characterization of the compounds by various methods, a new compound and three known compounds were first isolated from the *Cistanche* plant. A complete set of isolation, purification, and identification methods provides an important tool for the isolation of the active components of *Cistanche* plants. A reagent-saving, rapid, and feasible DPPH-UPLC-PDA method was developed to better exploit the *C. deserticola* resource [88]. The method can be used to correlate the chemical characteristics of *C. deserticola* with its biological activity without separation and purification and can be used for multicomponent analysis of active substances in other foods and food products. The study of new compounds could expand the potential applications of *Cistanche* and further inform the development and utilization of *Cistanche* as a functional food or supplement [89]. With the development of specific new adsorbent materials in the future, it will be possible to make the isolation and purification of the active ingredients of *Cistanche* accurate and fast.

Evidently, as research progresses, the study of *Cistanche* has gradually shifted from macroscopic to microscopic research at the molecular level. Nowadays, researchers are increasingly focusing on the specific mechanisms of *Cistanche* in disease treatment, and studies in this field are developing rapidly. Echinacoside is a natural PhGs in *Cistanche,* and growing studies have shown that echinacoside has good potential in inhibiting the progress of neurodegenerative disease. In 2018, Liu et al. published a review dedicated to echinacoside research [69]. Echinacoside is widely available in *C. deserticola*, *C. salsa*, *C. sinensis*, and *C. tubulosa*. Based on the abundant laboratory data, echinacoside shows a high degree of positive activity in neurological diseases and has broad therapeutic potential. Yet, at present, despite the extensive experimental evidence describing the promising pharmacological effects of echinacoside, clinical questions regarding the low bioavailability, extremely rapid metabolism, and the molecular targets of effect are still not well explored, and there have been limited clinical trials on the compound. Therefore, more clinical trials on the safety and the druggability of echinacoside are necessary. In 2022, Li et al. published another review on the neuroprotective effects of echinacoside [70]. Although clinical evidence is still lacking, a growing body of evidence suggests that echinacoside may become an effective and safe substance for the treatment of neurodegenerative diseases in the future due to its beneficial effects on neuronal function, including protection of mitochondrial function [90], anti-oxidative stress [91], anti-inflammatory [92], anti-endoplasmic reticulum stress [93], and regulation of autophagy [94]. Echinacoside has a wide range of neuropharmacological activities and is considered as a potential natural active ingredient with a broad spectrum and multi-target effect in the treatment of Alzheimer’s disease [20,39,95,96]. Studies have shown that the prevalence of Parkinson’s disease is expected to double in the next 20 years, as the global population ages [97]. Echinacoside can inhibit neuroinflammation involved in Parkinson’s disease progression through multiple pathways. Activation of microglia-mediated inflammatory response is a major component of the pathological process of Parkinson’s disease. In the echinacoside group of 1-methyl-4-phenyl-1,2,3,6-tetrahydropyridine (MPTP) Parkinson’s disease model mice, Iba-1, a specific marker of midbrain microglia, was decreased, and echinacoside treatment inhibited microglia activation, thereby improving brain inflammation [98]. It has also been reported that echinacoside improved neuropathological status in mice with Parkinson’s disease via neuroprotection and inhibition of activated microglia-mediated nucleotide-binding oligomerization domain (NOD)-like receptor protein 3 (NLRP3), Caspase (CASP)-1, and interleukin-1β (IL-1β) signals of inflammation [99]. These data suggest that the neuroprotective effects of *Cistanche* will remain a major topic of research in the future.

Anti-aging is also one of the hot topics in *Cistanche* study. It is believed in some regions of China and Japan that, beyond its conventional use, the daily consumption of *Cistanche* Herba is key for longevity [67]. *Cistanche* extract is widely used in Chinese medicine for its believed ability to promote immune function in the elderly. Animal experiments have shown a significant effect of *C. deserticola* extract in prolonging life span by antagonizing immune senescence [100]. An interesting study found that echinacoside increased the tolerance of *C. elegans* to heat shock and oxidative stress without any effect on pharyngeal pumping rate and production of offspring [90]. These effects of echinacoside were further demonstrated in a study where the supplementation of the compound promoted longevity and increased the stress response in C. elegans, via the modulation of the nuclear localization and transcriptional activity of *daf-16*, leading to the regulation of the transcriptional levels of *daf-16* target. These findings are of great significance to the use of echinacoside to improve the outcome of human aging-related diseases. The development of modern drugs, traditional Chinese medicine prescriptions, dietary supplements, and other related products has progressed significantly. In 2021, Song et al. published a review in *Medicinal Research Reviews* [24], in which the authors deeply and comprehensively summarized and discussed how *Cistanche* has been transformed from an endangered species to a big brand industry in the past decades. At the same time, the prospects and challenges of *Cistanche* as a medicinal and edible plant were explored. In recent years, an increasing number of *Cistanche*-related health products and Traditional Chinese medicine (TCM) prescriptions have entered the market. Nevertheless, some obvious bottlenecks still need to be overcome towards cultivating Cistanches into a big brand of Chinese medicine. These include (1) being parasitic plants, the need to clarify their parasitic mechanism; (2) in-depth product development; and (3) environmental balance for the cultivation of the plants. *Cistanche* has long been a valuable and precious medicinal herb, but the current limited supply is still a major factor limiting the development and promotion of Cistanches Herba related products. Going forward, the breeding and cultivation of *Cistanche* remain important research topics. It is believed that the cultivation yield will be further improved after the parasitic mechanism between *Cistanche* and its host is completely clarified.

## 4. Data and Methods

### 4.1. Data Sources and Search Strategy

Web of Science Core Collection (WOSCC, Clarivate Analytics, 1980-present) is the premier resource on the WOS platform, containing over 12,000 of the most impactful international journals in which articles are published under rigorous peer review [101]. Through the advanced search function of WOS, a subject search was performed in the core database as a string. The strategies used to retrieve were as follows: Topical Subject = *Cistanche* OR Title = *Cistanche* OR Author Keywords = *Cistanche* OR Abstract = *Cistanche*; the language type = English; and the document type = articles or reviews or book or book chapter. The literature was searched for the period 1980–2022 (data as of July 2022) and indexed in SCI-EXPANDED, SSCI, AHCI, CPCI-S, CPCI-SSH, BKCI-S, BKCI-SSH, and ESCI. Upon retrieval following the above method, a total of 443 records were included for quantitative analysis in CiteSpace, including 417 articles, 21 reviews, and 5 book chapters.

### 4.2. Research Tools and Visualization Methods

Microsoft Office Excel 2019 was used for quantitative analysis of the literature. Scientometric analysis was performed with CiteSpace 6.1.R3, including literature co-citation, publication countries, institutions, authors, journals, and keywords. The parameters of CiteSpace software had the following settings: the time slice was from 1980–2022, number of years per slice was 1; the sources of selected terms were title, abstract, author keywords, and keywords plus; the selection criteria g-index value was set to 25. The statistical indicators in the CiteSpace visual analysis included frequency and intermediary centrality. According to the different research content, the literature statistics tools of CiteSpace and WOS database were used to set different analysis parameters to visually analyze the research content.

## 5. Conclusions and Prospects

In this study, we analyzed the research output on *Cistanche* over the last 40 years and provided a visual and analytical review of the trends and progress in this field. It is clear that *Cistanche* study is an evolving field with an increasing number of publications, due to the increased awareness of *Cistanche* as a medicinal and edible herb. The concentration of the top 10 institutions and the top 25 authors in Asian countries (especially China) indicate their outstanding contributions to the advancement of *Cistanche* study. Furthermore, in the past decades, studies on *Cistanche* have mainly focused on their rich active substances and pharmacological effects. Breeding and cultivation of *Cistanche* has seen encouraging progress. The development and application of *Cistanche* as functional food is an emerging research hotspot. Our analysis provides intuitive and systematic knowledge on the status of *Cistanche* study for future guidance in related investigations.

The current study provides a rich and rigorous analysis of *Cistanche* by using bibiometric analysis. However, we analyzed only the papers written in English and within WOS. It may not be sufficiently comprehensive to reflect the entire research panorama. For example, we searched a major Chinese scientific literature database, China National Knowledge Infrastructure (CNKI), and retrieved more than 2000 *Cistanche*-related publications, which are not within the current analysis.

## Figures and Tables

**Figure 1 plants-12-01098-f001:**
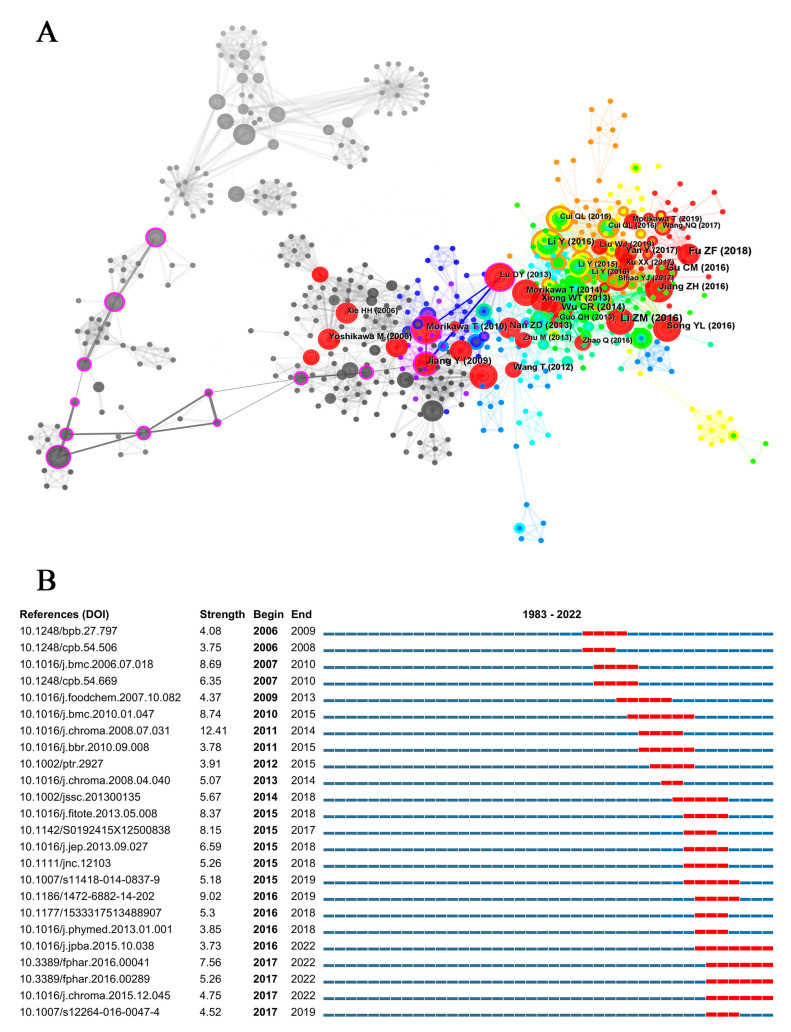
Reference analysis in *Cistanche*-related study. (**A**) Nodes in the network represent references. Node size represents the number of citations. Node color: average time to appear, color from white to red, time from 1980 to 2022. The purple circles represent publications with significant meaning. (**B**) Top 30 references with the strongest citation bursts in *Cistanche*-related study. The time period of the citation burst is indicated by the red line.

**Figure 2 plants-12-01098-f002:**
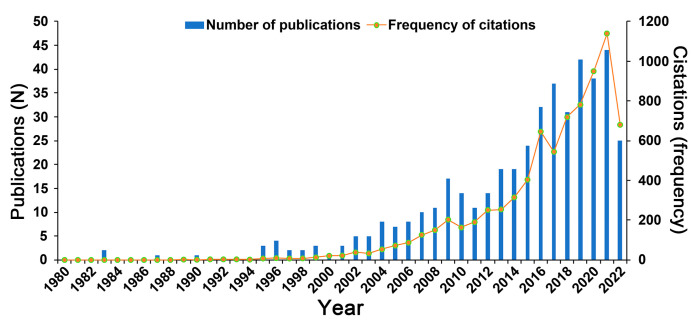
Number and citation frequency of publications in *Cistanche-*related study (1980–2022).

**Figure 3 plants-12-01098-f003:**
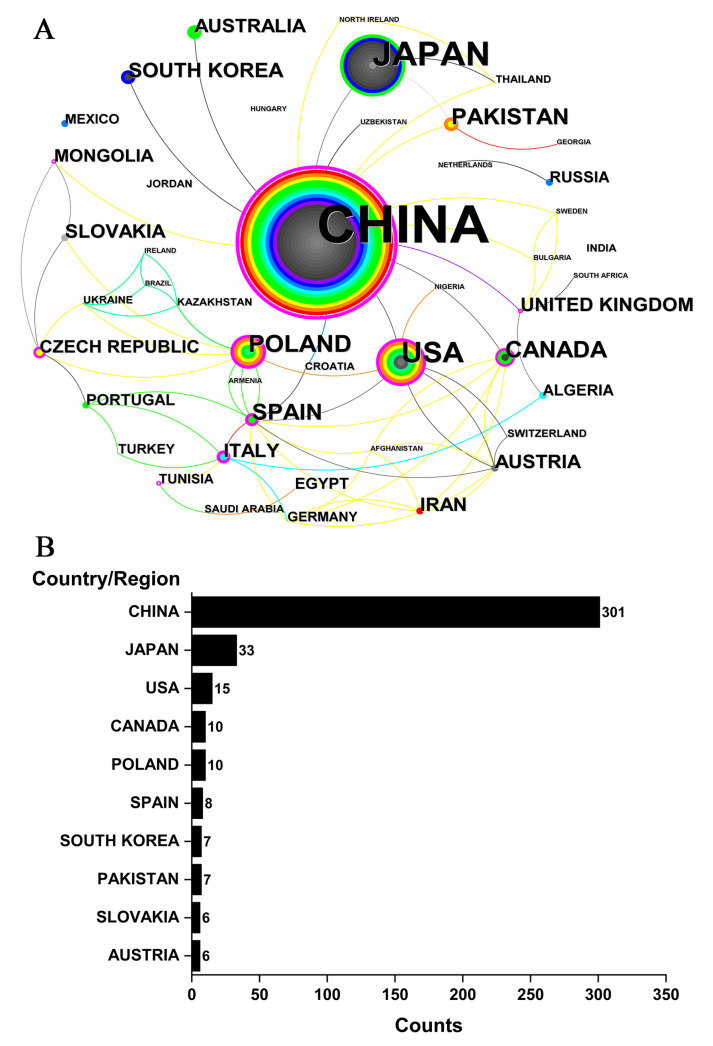
Country analysis of *Cistanche*-related study. (**A**) Node size represents the number of publications. Node color: average time to appear, color from white to red, time from 1980 to 2022. The purple circles indicate that these countries have a significant role in the study. (**B**) Top 10 countries with the most publications.

**Figure 4 plants-12-01098-f004:**
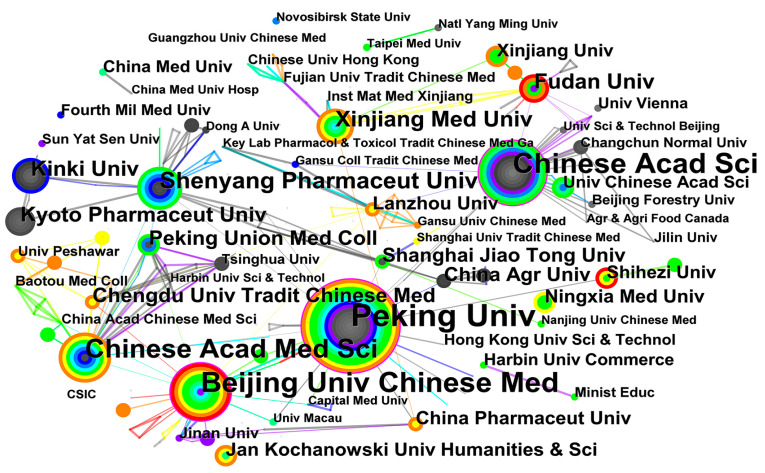
Institution collaboration network in *Cistanche*-related study. Node size represents the number of publications. Node color: average time to appear, color from white to red, time from 1980 to 2022. Purple circles indicate that these institutions have made significant contributions to the study.

**Figure 5 plants-12-01098-f005:**
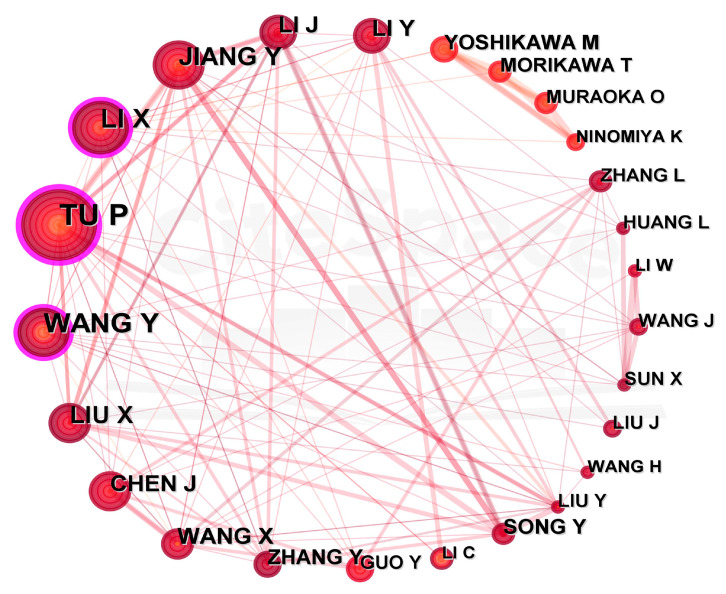
Top 25 authors in *Cistanche*-related study (1980–2022). Node size represents the number of publications. Purple circles indicate that these authors made significant contributions to the study. The diameter of lines represents the degree of cooperation.

**Figure 6 plants-12-01098-f006:**
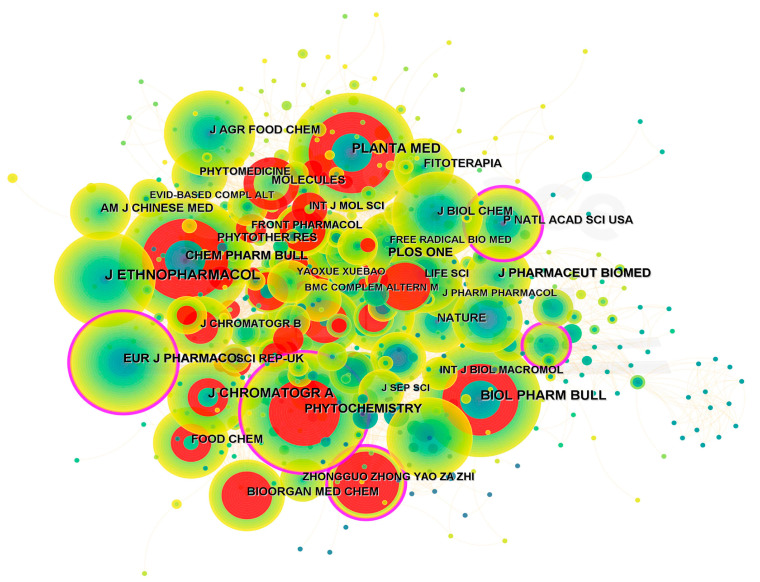
Co-cited journals in *Cistanche*-related study (1980–2022). Node size represents the number of citations. Purple circles indicate that the journal is a highlighted citation. Red nodes indicate journals have a citation burst over time.

**Figure 7 plants-12-01098-f007:**
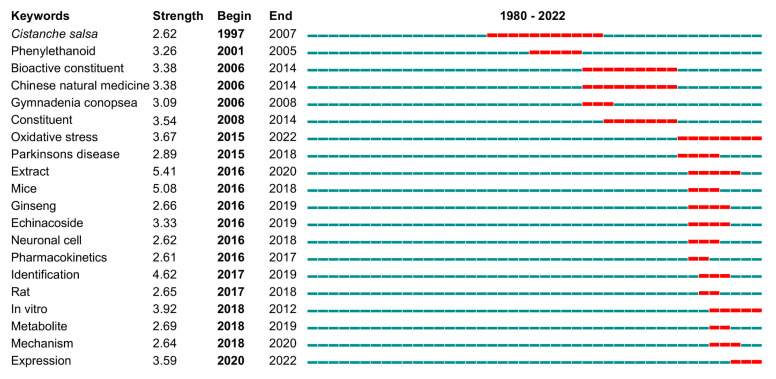
Top 20 keywords with the strongest citation bursts in *Cistanche*-related study (1980–2022). The time period of the citation burst is indicated by the red line.

**Figure 8 plants-12-01098-f008:**
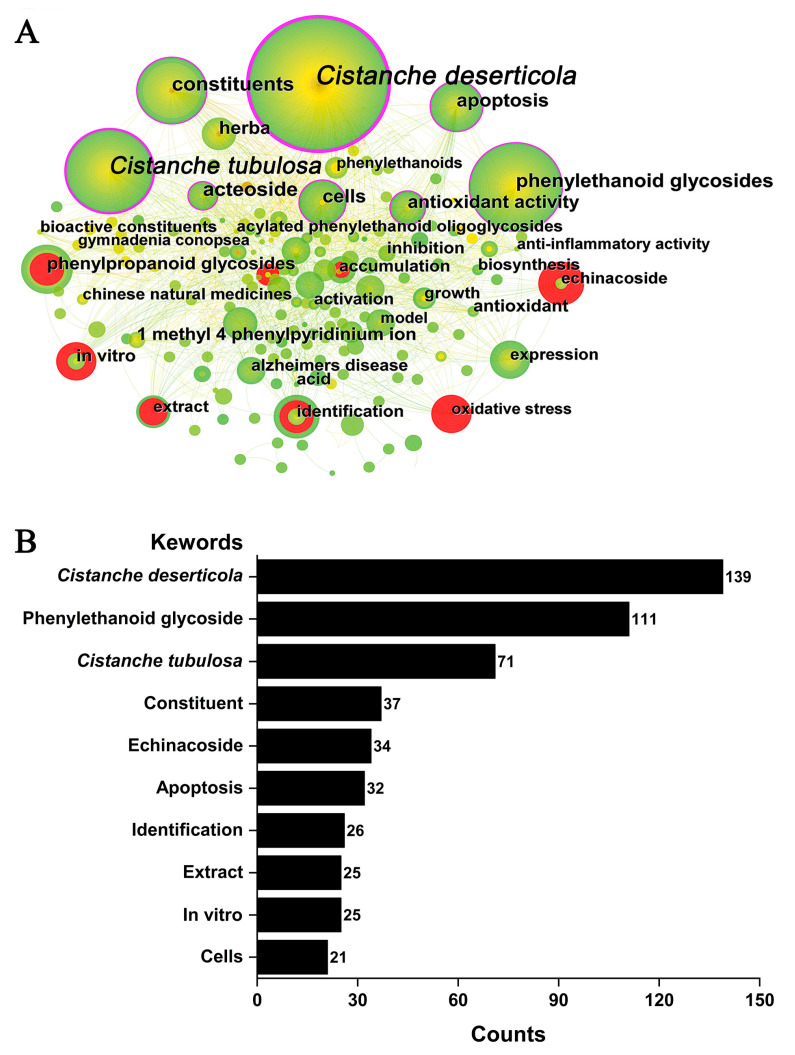
Keyword analysis in *Cistanche*-related study (1980–2022). (**A**) Node size represents the frequency of keyword occurrences. Purple circles indicate that the keyword is the focus of the study. Red nodes indicate hot spots of study. (**B**) The top 10 keywords with the most frequency.

**Table 3 plants-12-01098-t003:** Top 10 funding agencies.

Rank	Agency	Number of Records	Percentage (%)
1	National Natural Science Foundation of China Nsfc	154	34.76
2	Ministry Of Education Culture Sports Science and Technology Japan Mext	13	2.94
3	Ministry Of Science and Technology China	10	2.26
4	China Postdoctoral Science Foundation	7	1.58
5	Grants In Aid for Scientific Research Kakenhi	7	1.58
6	Japan Society for The Promotion of Science	7	1.58
7	National Key Technology R D Program	7	1.58
8	Quality Guarantee System of Chinese Herbal Medicines	7	1.58
9	Beijing Natural Science Foundation	6	1.35
10	Chinese Academy of Sciences	6	1.35

**Table 4 plants-12-01098-t004:** Keyword Cluster Analysis.

Cluster ID	Silhouette	Citation Year	Label (LLR)	Included Keywords
0	0.742	2011	Parkinsons disease	Parkinsons disease; endoplasmic reticulum stress; Orobanche genus; Chinese tonic herb; hepatoprotective action
1	0.844	2005	Cistanoside a	*Cistanche deserticola*; *Abeliophyllum distichum*; Chinese herbal medicine; deserticola; tissue
2	0.794	2012	Antioxidant activity	*Cistanche deserticola*; precursor feeding; comprehensive habitat suitability; 3-acetic acid; *Betula platyphylla*
3	0.906	2012	Phylogeny	Taxonomy; *Cistanche*; host; systematics; parasite
4	0.821	2011	Phenylalanine ammonia lyase (PAL)	*Cistanche deserticola*; total glycosides; permeability; mitochondrial precursor protein; antiviral agent
5	0.933	2007	*Cistanche tubulosa*	*Cistanche tubulosa*; phenylethanoid glycoside; *Abeliophyllum distichum; Rubia yunnanensis*; psoralea corylifolia
6	0.811	2007	*Cistanche deserticola*	*Cistanche deserticola*; agent; systemic fungal infection; *Candida albican*; flow cytometry
7	0.766	2016	*Cistanche* Herba	*Cistanche tubulosa*; upper limit; wide polarity span; simultaneous determination; interaction chromatography

## Data Availability

Data is contained within the manuscript.

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
