# Peer review of "Studies on Cistanches Herba: A Bibliometric Analysis"

_plants, 2023, doi:10.3390/plants12051098_

Round 1
Reviewer 1 Report
The paper quite well written, but there are doubts about whether the choice of journal is really appropriate. The paper presents the results of a purely scientometric study on a plant subject, but has little relevance to the plants themselves. I would therefore suggest that the authors consider publishing the article in another journal that is more appropriate to the subject matter.
1. The title of the article should avoid abbreviations that are unnecessary and even misleading (spp.). The abbreviation spp. (= species pluribus; multiple species) is used when the species of a particular genus are unknown or unidentified. The abbreviation sp. (= species; singular) may also be applied in the same sense. Since the authors clearly identify the species in the text of the paper, the abbreviation spp. is terminologically misleading. The same applies to the text of the article, where this abbreviation is often used, but interchangeably with the genus name without abbreviation (Cistanche). I would therefore suggest drawing attention to this terminological inaccuracy, which is becoming more and more widespread, but is causing confusion.
2. What is the purpose of the abstract (line 20) referring to the authors of this article? After all, the authorship of the article is already clearly indicated.
3. Why do the authors only give the translated name of the plant in English and Chinese, when the genus in discussion is found in Africa, the Mediterranean region and large parts of Asia, where plants have their own names. More national names of the genus should be given or only the scientific name should be used.
4. The authors categorise the article as a review article, but it is essentially scientometric, as mentioned above, and should contain the results of the scientometric study and a subsequent discussion. If this concept of the article were to be retained, it would still be necessary to discuss the obtained scientometric results.
5. If the article is still to be published in Plants, the introduction should be expanded. It should not only give a brief overview of the use of plants for medicinal or food purposes, but also of the resulting conservation problems. The intensive exploitation of the resources of plants that cannot be cultivated under artificial conditions, as the authors point out (semi-wild; lines 434-435; however, see lines 468-469), inevitably puts species populations at risk.
6. Why do the authors separate part of the scientometric component into Chapter 4? In my opinion, subsections 4.1 and 4.2.1 could form a separate chapter. From section 4.2.2 onwards, the review of the publications is already underway and this part could be in a separate chapter. After all, there is no limit to the number of chapters and subsections in review articles.
7. In my opinion, the style of the review is unusual. The article does not analyse or compare the claims made in other references, but merely reports on who has researched what, what methods have been applied and what has been found, without any critical assessment (from 4.2.2 onwards). Critical evaluation of published results is an important feature of review articles, and this is the primary purpose of such articles. If the authors were to keep to the current concept of the article, then it should be classified as a scientometric study and should have the components of a research article.
8. The list of references has been prepared in complete disregard of the requirements of the journal. What is most striking is that the bibliography of all the articles does not include the year of publication, nor does it include a reference to the DOI (although this is not necessary). The absence of the year of publication makes it extremely difficult to find the cited references. To be honest, I have never seen this style of citing literature sources before.
9. There are technical errors in the article, e.g. (a) Haloxylon Ammodendron should be Haloxylon ammodendron (the species epithet is never capitalised); (b) the genus name at the beginning of the sentence should be written in full, not shortened to C. (= Cistanche).
I fully understand that authors may have different opinions and visions on the structure of an article, and that there may be different ways of presenting the information, but all of them must be based on solid logic. Even very unconventional ways of presenting the information are possible, but the principles should be clear to any potential reader, and these can always be discussed in the introduction, in the aims and objectives, or in the methodology section.
Reviewer 2 Report
The paper is related to the information associated to the study of a plant using data bases. Need to improve some aspects: key words in alphabetical order, the references need to be written with journal names complete or abbreviated.
Including some figures related to the important function of the plant must help to understand
Reviewer 3 Report
The manuscript plants-2081738-peer-review-v1 “Cistanche spp., a popularized herb with increasing health benefit and diverse applications: An updated review and scientometric analysis of literature” presents interesting data on the bibliometric analyses of research on Cistanche spp. articles published from 1980 to 2022.
If I correctly read the paper, Cistanche spp. (the so-called desert ginseng or Rou Cong Rong in Chinese) are holoparasitic plants in the family Orobanchaceae that parasitize Tamarix spp. and Haloxylon ammodendron. Along with the members of the genus, Cistanche deserticola, Cistanche tubulosa, Cistanche salsa and Cistanche sinensis are the primary source of the Chinese herbal medicine cistanche. “Cistanche herbs” is known for their broad medicinal functions, especially in hormone regulation, anti-ageing, anti-dementia, antitumor, anti-oxidative, neuroprotective, and hepatoprotective effects. Published articles were searched in the Web of Science database from 1980 to 2022 using ‘Cistanche’ in Topical Subject, Title, Keywords and Abstract, ‘English’ as language type, and ‘articles or reviews or book or book chapter’ as the document type. The recovered 443 papers (422 articles and 21 reviews) were mapped using Microsoft office Excel 2019 and the classic CiteSpace software. The analysis identifies Countries, institutions, journals, authors, keywords, the most cited literature, and the greatest number of publications through different metrological means. The results revealed a significant increase in research outputs on this topic year by year. During the period 1980-2022, 46 countries/regions, 330 institutions, and 536 authors were involved in this field. China published 67.9% of selected papers and revealed a close collaboration. Peking University was the main publishing institution, while Pengfei Tu was the author with the highest number of publications. Cistanche deserticola and Cistanche tubulosa were highlighted in the research. Active substances composition, pharmacological effects, breeding, and cultivation were mainly focused on. The application as a functional food is an emerging research of Cistanche spp.
Notwithstanding the scientific sound of this study, the presentation in the form of a manuscript is exposed in a cumbersome way and requires several adjustments.
Follow the format of “Plants” to rewrite the manuscript.
Pay attention to the layout of the work: legends of Figure 1, Figure 2, and table 1 layout are divided into two consecutive pages.
Arrange the References section following the “Plants” guidelines.
The paper presents two different hybrid materials. The first part is a research-article-type, while the second part is a review.
To meet the requirements of a scientific journal, it would be necessary to move the 4.2.1 and 4.2.2 sections as an introduction and focus the research on bibliometric analyses.
Some suggestions to improve the manuscript.
Line 2: What does “popularized” mean?
The title is not pertinent to the topics of the manuscript. A suggestion:
Studies on holoparasitic Cistanche spp.: bibliometric analysis and visualization.
I think that the abstract is a very important part of the manuscript, after reading the manuscript I suggest rewriting it.
Improve the “Introduction” section and report information from the 4.2.1 and 4.2.2 sections.
A description of Cistanche species could help the readers.
Using the following subdivision:
Line 61: use “2. Date and Methods”
Line 62: use “2.1. Data Sources and Search Strategy”
Line 76: use “2.2. Research tools and Visualization methods”
Line 95: use “3. Results”
3.1 Analysis of important literature
3.2 Annual number of publications
3.3 Main Research Forces
3.3.1 Major Countries
3.3.2 Main Institutions
3.3.3 Main authors
3.4 Source Analysis of Journals
3.5 Hotspot Analysis
3.5.1 Outbreak Word Analysis
3.5.2 Keyword analysis
4. Discussion
5. Conclusions and Prospective
Figure 1:
1A: Insert axis categories
1B: Improve magnification
I suggest three different figures
Improve the legend
Figure 2
2A and 2B: Improve magnification
I suggest three different figures
Improve the legend
Figure 3
3A and 3B: Improve magnification
I suggest three different figures
Improve the legend
What does “Table 1” mean?
Figure 4
4A and 4B: Improve magnification
I suggest three different figures
Improve the legend
Line 473: What does “Summary” mean? Prefer “Conclusions”
Round 2
Reviewer 1 Report
The revised version of the paper has been significantly improved following peer reviews and now meets the structural requirements for articles. Having read the content of the article, I would like to make a few editorial and technical comments, as well as comments on the use of certain terms.
1. Line 46. The genus Cistanche is not distributed worldwide, but only in Europe, Asia and Africa. There are no species of the genus on the American continents and Australia.
2. In Figure 3(a) I would suggest to write only China, because other countries are also given short forms of their names, not their official (long) names. In addition, England should be replaced by United Kingdom.
3. Why are the names of some countries in Figure 3(b) capitalised and others in normal typeface? If there are differences, these should be explained or uniformity should be maintained.
4. Some of the entries in Figure 4 are illegible even if the image is enlarged significantly. Technical solutions to the problem should be sought.
5. In Figure 7, on the y-axis, the names of the genera should be written according to the nomenclature rules. Generic names are capitalized always.
6. In Figure 8, the comments are the same as for Figure 7. Shouldn't 'deserticola' be combined with 'Cistanche deserticola'?
7. Line 418. The family Orobanchaceae cannot include other families. Subject error, sentence must be corrected. Suggest: [...] including all members of the Orobanchaceae family [...].
8. Line 511 onwards. I suggest that items (a), (b) and (c) be written (a), (b), etc., or numbered (i), (ii), etc., or (1), (2), etc.
9. I suggest that the editing of the text should be entrusted to a professional editor.
Reviewer 2 Report
The authors attend the observations , however still missing inormation about specific uses of the plant
Reviewer 3 Report
The revised version of the manuscript plants-2081738-peer-review-v2 “Studies on holoparasitic Cistanche: bibliometric analysis and visualization”, was improved, but it remains in some parts difficult to read, repetitive and confusing.
The paper does not reach the standard of Plants.
Some suggestions and adjustments.
In this version, the word “Cistanche” indicates Chinese herbal medicine, then avoid italics and “holoparasitic” in the title.
Improve the abstract.
Line 23: What does “analyses” mean?
Line 30: What does “literatures” mean?
Line 49: use “the Iberian” instead of “Iberian”
Line 59: use “Cistanche Herba plants” instead of “Cistanche”
Line 61: If “Phenylethanoid glycosides (PhGs)” indicate two or more, use “are” instead of “is”
Lines 62-138: The periods are confused. Rewrite.
Line 94: use “increasing” instead of “growing”.
Line 97: What does “Cistanche” mean?
Line 97: What does “its” mean?
Line 106: What does “Phytochemical” mean?
Lines 140-168: improve.
Line 178: use “12,000 of” instead of “12,000”.
Line 191: use “Research” instead of “Rearch”.
Line 192: use “Office” instead of “office”.
Lines 200-205: improve.
Figures must be self-explanatory and easy to understand without any extra explanation. Improve all figure legends.
Figure 1:
1B: Improve magnification.
Eventually, use the Doi alone to indicate the reference.
Or report essential data as a table.
Tables must be self-explanatory, clear, and easy to understand without any extra explanation.
Table 1 is unreadable.
Figure 2:
Use “Publications (N)” instead of “Publications”
Use “Citation (frequency)” instead of “Citations”
Arrange the References section following the “Plants” guidelines (https://www.mdpi.com/journal/plants/instructions). In particular: “Include the digital object identifier (DOI) for all references where available”.
When sending the correct file, a highlighted version is preferable to the review mode.
Round 3
Reviewer 3 Report
According to the corrections that have been made, the manuscript Studies on Cistanches Herba: bibliometric analysis and visualization can be accepted after minor adjustment.
Lines 45-48: rewrite
Lines 124-125: use ‘seed germination and haustorium production’ instead of “seed and haustorium germination”
Lines 330-332: rewrite
Arrange References 2,4, 7, 16, 20-21,24-27,31, 37, 39-43, 14-57, 60-61, 64, 66-67, 81, 83, 86, 88-89, 97, 100,103 following the “Plants” format.
